# From deconfined spinons to coherent magnons in an antiferromagnetic Heisenberg chain with long range interactions

**Luhang Yang and Adrian E. Feiguin⋆**

Department of Physics, Northeastern University, Boston, Massachusetts 02115, USA

⋆ a.feiguin@northeastern.edu

## Abstract

We study the nature of the excitations of an antiferromagnetic (AFM) Heisenberg chain with staggered long range interactions using the time-dependent density matrix renormalization group method and by means of a multi-spinon approximation. The chain undergoes true symmetry breaking and develops long range order, transitioning from a gapless spin liquid to a gapless ordered AFM phase. The spin dynamic structure factor shows that the emergence of Néel order can be associated to the formation of bound states of spinons that become coherent magnons. The quasiparticle band leaks out from the two-spinon continuum that is pushed up to higher energies. Our physical picture is also supported by an analysis of the behavior of the excitations in real-time.



## 1 Introduction

Heisenberg antiferromagnets (AFM) provide a testbed for spin-wave theory [1, 2]. However, it is well known that the spin-wave approximation is not a good starting point to describe

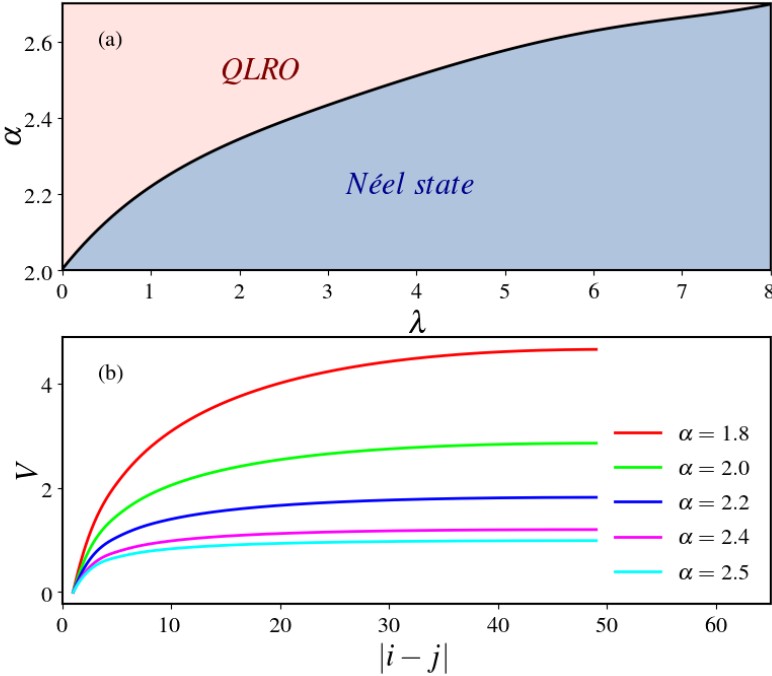

Figure 1: (a) Phase diagram of the Heisenberg chain with long range staggered interactions reproduced from Ref. [24], as a function of the coefficient $\lambda$ and exponent $\alpha$; (b) confining potential as a function of $\alpha$ for $\lambda = 1$.

the dynamics of spin chains. Instead, their spectrum is determined by propagating domain walls (spinons), which are the natural basis of excitations in one spatial dimension [3–6]. The fractionalization of excitations is an exotic many-body phenomenon that has mobilized both experimentalists and theorists for decades and has been observed in 1D spin chains and ladders [7–18].

Spin-chains are not well ordered antiferromagnets: their correlations decay algebraically and they do not develop true long-range order. Higher dimensional magnets may develop long-range order and, in such a case, excitations are gapless magnons with well-defined Goldstone modes. Spinons, or fractionalized excitations, are not only a feature of 1D spin chains but are also expected to emerge in 2D spin liquids – states that do not break continuous symmetries – as postulated by the theory of deconfined quantum criticality [19–21]. To reconcile these two pictures we interpret magnons (which carry spin 1) as bound states of spinons (that carry spin 1/2). Notice , however, that in 1D and even in 2D spin liquids [22], it is possible to have bound states of spinons without long range order. In such a case, these excitations are referred to as "triplons" (the simplest example is a triplet excitation on top of a dimerized gapped valence bond solid) [23].

On the 2D square lattice, a prototypical antiferromagnet, the spin wave dispersion agrees with numerical results with high accuracy in the entire Brillouin zone, with the only deviations along the $(\pi, 0) - (\pi/2, \pi/2)$ path [25, 26]. Along this segment, the spin-wave theory dispersion is essentially flat, while numerical results indicate a dip. Recent experiments are in excellent agreement with numerics [27–29] which have prompted the speculation of physics beyond magnons. In particular, in recent low- temperature polarized neutron scattering experiments [29] , the broad and spin-isotropic continuum in $S^z(k, \omega)$ at $q = (\pi, 0)$ was interpreted

as a sign of deconfinement of spinons in a region of momentum space. Recent numerical Monte Carlo results show, however, that magnons are still present and, even though their spectral weight may become very small, they never vanishes [30].

Similarly, neutron scattering experiments on the 2D triangular antiferromagnet $Ba_3CoSb_2O_9$ [31, 32] indicate that the spectrum consists not only of low energy magnon branches, but also high energy continua with a separation of the order of the exchange interaction $J$. The disagreement with expectations from spin-wave theory stimulated further theoretical work [33, 34] that pointed at deconfined spinons being the culprits of the high-energy features, with magnons consisting of bound states of spinons.

The above considerations beg the questions: how do gapless spinons evolve into gapless magnons and singularities in the spectrum into coherent Goldstone modes as we transition from a gapless spin liquid into a gapless antiferromagnet with long range order? [35, 36] In order to address these issues we resort to one dimensional spin-1/2 chains with staggered $SU(2)$-symmetric long-range interactions that allow us to realize actual spontaneous symmetry breaking and true antiferromagnetic order: The general problem can be formulated by means of the following Hamiltonian:

$$H = J \sum_i \vec{S}_i \cdot \vec{S}_{i+1} - \lambda J \sum_{|i-j|>1} \frac{(-1)^{i-j}}{|i-j|^\alpha} \vec{S}_i \cdot \vec{S}_j. \tag{1}$$

The long range nature of the interactions artificially increases the dimensionality of the problem and circumvents the restrictions imposed by the Mermin-Wagner theorem. At the same time, they introduce volume-law entanglement, making the calculations more challenging. However, the staggered phase enhances antiferromagnetism and avoids frustration, also making it amenable to quantum Monte Carlo calculations [24, 37, 38].

The phase diagram of the extended Heisenberg chain as a function of the coupling $\lambda$ and exponent $\alpha$ was obtained by Laflorencie *et al* in Ref. [24] using quantum Monte Carlo, who found a critical line separating Néel ordered and disordered phases with dynamic critical exponent $z < 1$ (see Fig.1(a)). This indicates that the system generally does not admit a description in terms of conformal field theory (which is not surprising), which should be manifested in the finite-size scaling of the spin gap and the curvature of the dispersion (which is sublinear), as well as the behavior of the entanglement entropy. Therefore, the chain undergoes a transition from a gapless ordered phase with strong AFM correlations, to a gapless disordered one with fractionalized excitations. The two regimes are characterized by an order parameter, the staggered magnetization $m = S^z(k = \pi)$, and by the nature of the excitations that should be reflected in the spectrum, given by its dynamic spin structure factor $S^z(k, \omega)$.

In this work, we focus on the particular case of $\lambda = 1$. In this limit, the Hamiltonian becomes:

$$H = -J \sum_{i \neq j} \frac{(-1)^{i-j}}{|i-j|^\alpha} \vec{S}_i \cdot \vec{S}_j. \tag{2}$$

The manuscript is organized as follows: in section 2 we discuss the spectral function obtained by means of the time-dependent density matrix renormalization group method (tDMRG) [39–42]. In section 3, we describe a multi-spinon analytical approach and analyze the results from the point of view of confined spinon excitations. We provide an intuitive picture of the nature of the excitations by studying their behavior in real time in section 4 and we finally conclude with a summary and discussion.

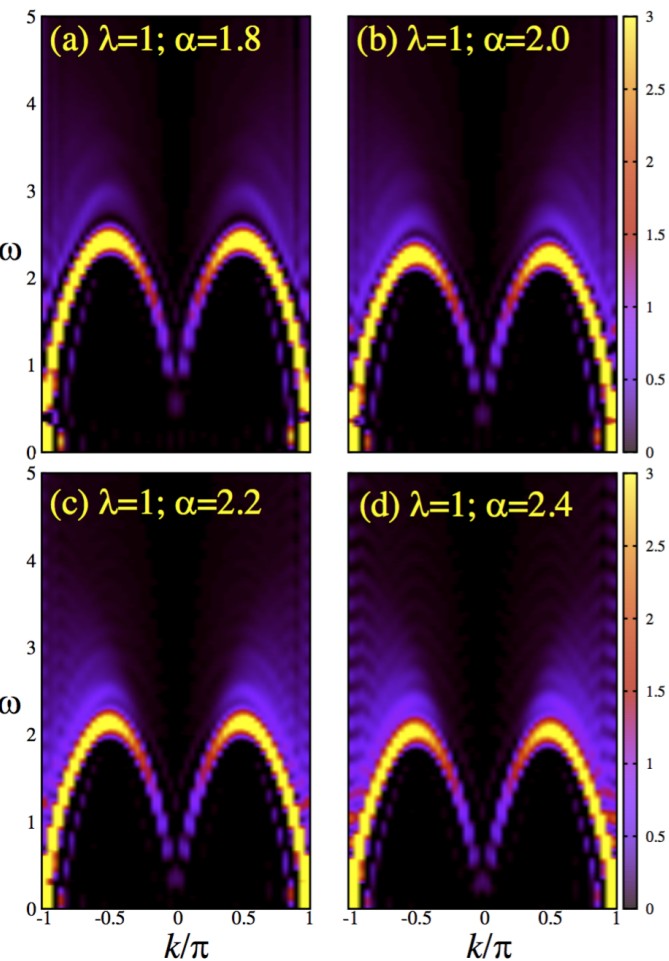

Figure 2: Momentum resolved dynamic structure factor $S^z(k, \omega)$ of the Heisenberg chain with long range interactions, $\lambda = 1$, and different values of $\alpha$ across the phase transition.

## 2 Spin dynamics

In the disordered phase of the model Eq.(2), excitations are described in terms of deconfined spinons. Assuming a spinon dispersion $\epsilon(k)$, the two-spinon continuum is constructed by all possible energies $\epsilon_2(k) = \epsilon(k_1) + \epsilon(k_2)$, with $k = k_1 + k_2$. For the conventional nearest-neighbor Heisenberg chain, the resulting spectrum is bounded from below by the des Cloizeaux-Pearson dispersion $\pi J/2|\sin k|$ [43], and the upper boundary of the continuum is $\pi J|\sin(k/2)|$ [44]. Therefore, it will be characterized by singularities and will not realize coherent quasiparticles, that in the spectrum would appear as $\delta$-like peaks, accompanied by incoherent background at high energies (that can correspond to spinons or a two-magnon continuum). Magnons are associated to symmetry breaking and the emergence of gapless Goldstone modes after some gapped mode condenses. However, it is possible to transition from a gapless phase without long range order (a gapless spin liquid) to an ordered one with a well defined order parameter. In this case, it is expected that the gapless deconfined excitations of the spin liquid will form bound states in the ordered phase. For this to happen, an attractive confining potential should be strong enough to overcome the kinetic energy of the free spinons. This is precisely what occurs in our model.

In the conventional 1D Heisenberg model, flipping a spin would create two domain walls.

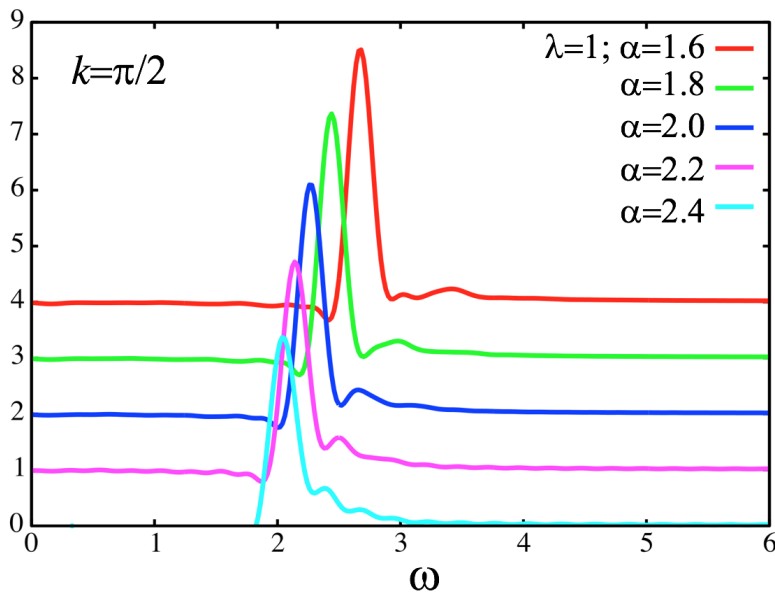

Figure 3: Spin dynamic structure factor across $k = \pi/2$ cuts for $\lambda = 1$ and different values of $\alpha$. The emergence of a sharp isolated peak leaking out of the continuum is clearly observed as $\alpha$ decreases and long range order is developed. The curves are shifted for clarity. Negative values are artifacts of the Fourier transform, as described in the text. The finite width of the peaks is determined by the maximum simulation time.

The energy cost of such excitation does not scale with the separation between the particles. However, in the case of long range staggered interactions, all spins interact with each other and the local disturbance is felt by the bulk of the chain. Separating the domain walls costs an extensive amount of energy, as depicted in Fig. 1(b), where we show the confining potential for different values of $\alpha$ and $\lambda = 1$, defined as the energy cost of moving two domain walls a distance $r \geq 1$ apart:

$$V(r) = E(r) - E(1), \tag{3}$$

where $E(r)$ is calculated in the Ising limit as:

$$E(r) = \sum_{i \neq j} (-1)^{i-j+1} \frac{\langle 0, r | S_i^z S_j^z | 0, r \rangle}{|i-j|^\alpha}, \tag{4}$$

and the state $|i, j\rangle$ represents two spinons at positions $i, j$ (*e.g.* as in Fig. 4(c)).

We are interested in determining signatures of confinement in the excitation spectrum of the model. In order to obtain the spin dynamic structure factor we used the time-dependent DMRG method (tDMRG) [39–42] following the prescription detailed in the original work Ref. [39]. The idea consists of calculating the two-time spin-spin correlator:

$$\langle S_r^z(t) S_0^z(0) \rangle = \langle \psi_0 | e^{iHt} S_r^z e^{-iHt} S_0^z | \psi_0 \rangle, \tag{5}$$

where $S_0^z$ here is defined at the center of the chain, and $r$ is the distance from center. Fourier transforming to momentum space and frequency, we reconstruct the momentum resolved spectral function. The Fourier transform is carried out over a finite time range (in our case $t_{max} = 20$), which requires the use of a windowing technique to attenuate artificial ringing

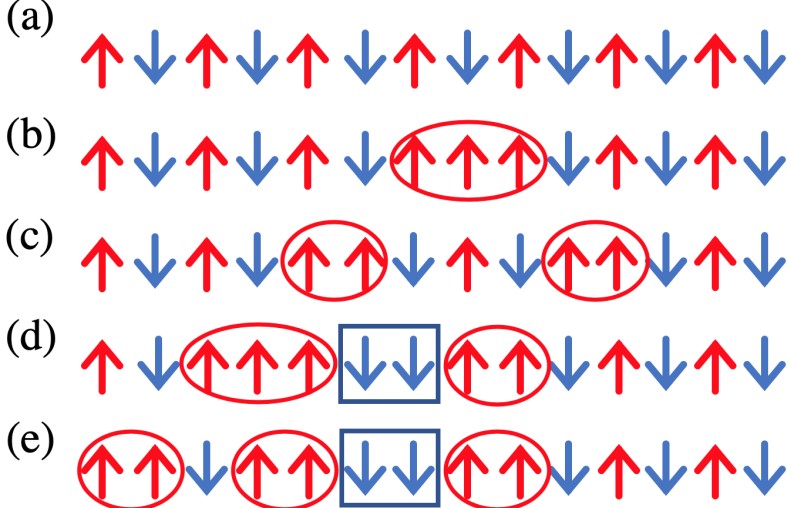

Figure 4: Possible configurations allowed in the variational Villain-like approach used in this work: a) Ising ground state; b) two confined spinons created after flipping one spin; c) two deconfined spinons; d) two confined spinons, one separate spinon and one anti-spinon; e) three spinons and one anti-spinon.

(satellite oscillations associated to the natural frequencies that lead to artifacts, such as negative values). The spectrum will exhibit an artificial broadening that is inversely proportional to the width of our time window. In addition, good resolution at high frequencies can be improved by using a small time-step, while a long time-window is necessary to improve resolution at low frequencies. In order to time-evolve the wave function, we use a time-step targeting procedure with a Krylov expansion of the time-evolution operator [45] and a time step $\delta t = 0.05$ (time is measured in units of $J^{-1}$ and $J$ is our unit of energy).

We study chains of length $L = 48$ using 400 DMRG states that guarantees that the truncation error remains smaller than $10^{-6}$ over the time window. Results for the dynamic structure factor are displayed in Fig.2 for $\lambda = 1$ and different values of $\alpha$ across the phase transition. The spectrum is bounded from below by a sharp peak, which for large $\alpha > 2.2$ corresponds to the edge of the two spinon continuum. For smaller $\alpha$, as we cross over to the ordered phase, the peak splits out from the continuum, that moves to higher energies. This can be seen more clearly in Fig.3, where we show cuts along the $k = \pi/2$ direction. The splitting of the peak and the shifting of the continuum to higher energies are signatures of the formation of bound states, which become coherent quasi-particles in the symmetry broken phase.

## 3  Two-spinon bound states

In order to develop intuition on the nature of the excitations and the confining mechanism, we study a related toy problem that will serve as a close approximation to the present situation. We will follow Villain [46] and assume that the ground state of the system has uni-axial symmetry (the Ising limit), and consider the dynamics of mobile domain walls. This is done by considering the motion of the spinons by means of spin flips, ignoring the action of these terms on pairs of spins that do not involve domain walls. The procedure was clearly outlined in Ref. [47] but, in our case, we need to consider the long-range nature of the interactions (a similar procedure was carried out for the ferromagnetic case in Refs. [48–51]). We firstly define the space spanned by all the possible configurations with two spinons in an antifer-

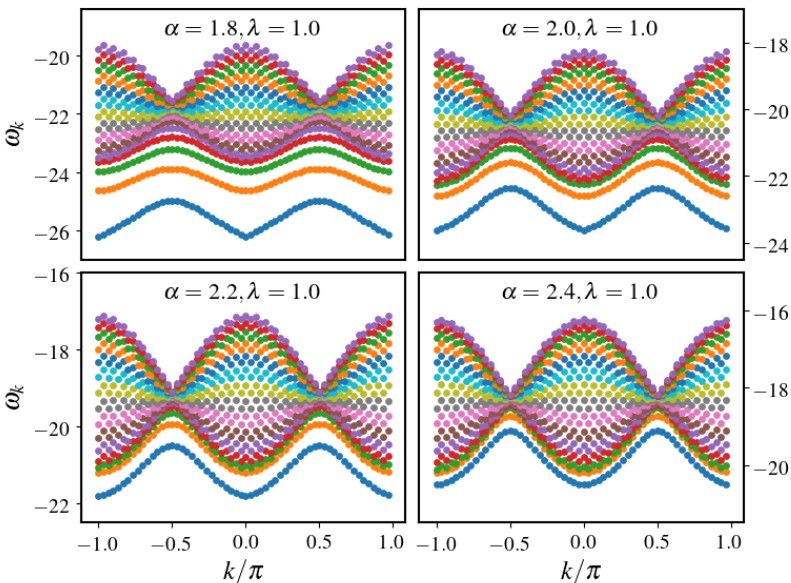

Figure 5: Two-spinon spectra for $\lambda = 1$ and different values of $\alpha$ obtained by using Villain's approach for $L = 60$, as described in the text.

romagnetic background $|i, j\rangle$, where $i, j$ are the positions of the domain walls, illustrated by Fig.4(a), (b) and (c). We then exploit translational symmetry and define wave functions in a sector with momentum $k = 2\pi n/L$ ($n = 0, \cdots L - 1$):

$$|\Psi(k, r)\rangle = \frac{1}{\sqrt{L}} \sum_{d=0}^{L-1} e^{ikd} T_d |0, r\rangle, \tag{6}$$

where the translation operator acts as $T_d |i, j\rangle = |i+d, j+d\rangle$ (periodic boundary conditions enforce position to be defined mod $(L)$). The Hamiltonian matrix elements are easily calculated and each momentum sector is diagonalized independently.

Results for the two spinon excitation spectrum for chains of length $L = 60$ are shown in Fig.5 for $\lambda = 1$ and several values of $\alpha$. We plot the energy of the two spinon state $E(k)$. For small $\alpha$ we see several bound states leaking out of the two spinon continuum. As $\alpha$ increases above the expected value for the transition in the isotropic case $\alpha_c \simeq 2.2$, the continuum tends to collapse and merge with the two-spinon bound states. This scenario agrees qualitatively with the observed behavior in the $SU(2)$ symmetric case. For small $\alpha$, the two spinon continuum is pushed to higher energies, and is clearly separated from the "magnon" band.

So far, our approach has ignored other possibilities that can be realized through long range spin flips. In fact, it is easy to convince oneself that two-spinon configurations can only propagate via nearest-neighbor spin-flips. In order to account for the long-range off-diagonal processes we have to consider states with three spinons and one anti-spinon, as illustrated in Fig.4(d) and (e). As a matter of fact, long range spin flips can create a proliferation of spinons and anti-spinons throughout the chain, but this is prevented by energetic considerations. For this reason and due to the fast growth of the number of possible configurations (scaling as $\sim L^3$), we only preserve those with one anti-spinon, and even so we can solve for sizes up to $L = 36$. The spectrum is now modified as seen in Fig.6, but the low energy features remain qualitatively similar. However, we see a continuum of high energy states corresponding to the new sector that gets mixed with the two-spinon continuum. A significant difference that we

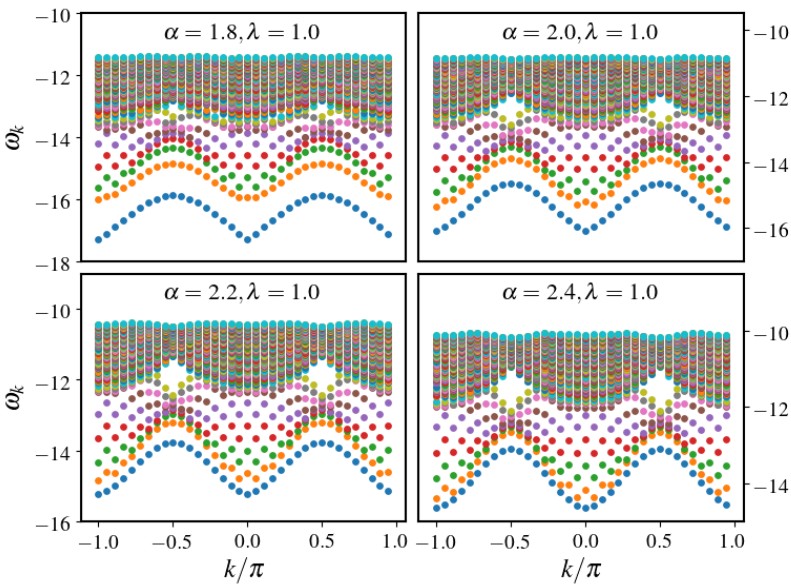

Figure 6: Spectra for $\lambda = 1$ and different values of $\alpha$ obtained by using Villain's approach for $L = 36$, including the sector with three spinons and one anti-spinon.

observe between these results and those in Fig.5 is that the number of bands leaking out of the continuum is suppressed. It is reasonable to expect that as more spinon and anti-spinon states are included, more states will appear at low energies. The fact that only one band finally survives would indicate that higher energy bound states (magnons) tend to decay into the continuum of multi-spinon excitations. Clearly, the low-energy physics is well described as consisting of bound states of spinons but, as we shall see, accounting for the additional spin fluctuations becomes important when it comes to understanding the real-time evolution of the system.

## 4   Real-time evolution

It is natural to ask whether the spinon confinement can be identified in a numerical "time-of-flight" experiment, in which a spinon is created at the center of a chain by the application of the $S^+$ operator, and left to evolve under the action of the Hamiltonian. Results obtained with tDMRG as shown in Fig.7 where we plot the correlations $\langle N_\uparrow(r, t)N_\uparrow(r + 1, t)\rangle$, where $N_\uparrow = (2S^z + 1)$. Notice that bound states are not necessarily localized in nearest neighbors, but actually are extended objects that have a characteristic size [37] that gets smaller with decreasing $\alpha$. However, to a certain extent, these correlations can help us develop intuition and, for this purpose, we shall refer to them as the "spinon density". Without long-range interactions, spinons propagate ballistically [52,53] and this is seen in Fig.7 as a "lightcone" with a velocity determined by the maximum slope of the spinon dispersion $v = \pi/2$. As the value of alpha decreases, spinons become more and more confined and, consequently, "heavier" (or slower), since bound states of spinons move coherently through second-order processes by means of two consecutive spin flips. However, a side effect of the long-range interactions is that spinons can "hop" longer distances. As a consequence, the originally sharp edges of the lightcone become more diffuse and, moreover, they acquire an apparent curvature that gives the misleading impression of an underlying "accelleration". As it turns out, this

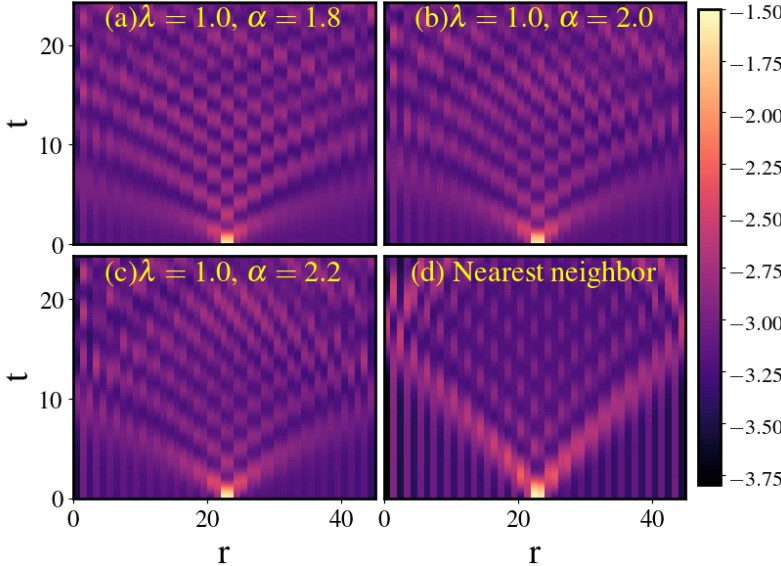

Figure 7: Domain wall expansion for $\lambda = 1$ and different values of $\alpha$, obtained with tDMRG for a chain with $L = 48$ spins. Results for the nearest neighbor case are also included. We show the "spinons density": $\langle N_\uparrow(r,t)N_\uparrow(r+1,t)\rangle$. Color density is in a log scale.

illusion is due to the superposition of two characteristic velocities, as we discuss next.

Our first attempt to explain the observed behavior is to use the Villain approximation with two spinons, presented in Fig.8. As we mentioned earlier, the long range interactions in this case enter only as a diagonal contribution, since long range spin flips produce a proliferation of spinons that take us outside of the two-spinon sector. However, we can already identify very interesting features as we change the exponent from basically $\alpha \to \infty$ (nearest neighbor interactions, only), to $\alpha = 1.8$. In the former case, we see a coherent ballistic propagation of the spinons with a well defined characteristic velocity, as expected, since the problem is equivalent to two non-interacting particles. For $\alpha = 2.2$ we identify two coexisting lightcones: a fainter one preserves the same slope as the free deconfined spinons, while the second one, with larger weight, describes coherent particles moving with roughly half the spinon velocity. It only makes sense to attribute these features to a bound state of spinons, a "magnon". As we reduce $\alpha$ even further, the free spinon lightcone loses weight, which is transferred to the magnons. For $\alpha = 1.8$ we can clearly identify a single magnon lightcone, as free spinons move to higher energies. In order to support these observations we also calculate $\langle N_\uparrow(r,t)N_\uparrow(r+1,t)N_\uparrow(r+2,t)\rangle$, or "magnon density", in Fig.9. In the nearest-neighbor limit we only see a very faint feature that loses weight as time evolves: the original flipped spin creates a state like the one depicted in Fig.4(b), but it is short lived and breaks into two spinons. As $\alpha$ decreases, the magnon lightcone becomes more and more coherent and correlates exactly with the features observed in Fig.8.

Having determined the coexistence of deconfined spinons and magnons in the lightcone, it rests to explain the apparent curvature in the DMRG results. For this, we need to extend our treatment by considering the possibility of three spinons and one anti-spinon to account for the long-range spin flips. The results for the spinon and magnon densities are presented in Fig.10 and Fig.11, respectively. We basically observe a "fan" or excitations covering the region between the magnon and free spinon wavefronts. Moreover, attempting to identify a charac-

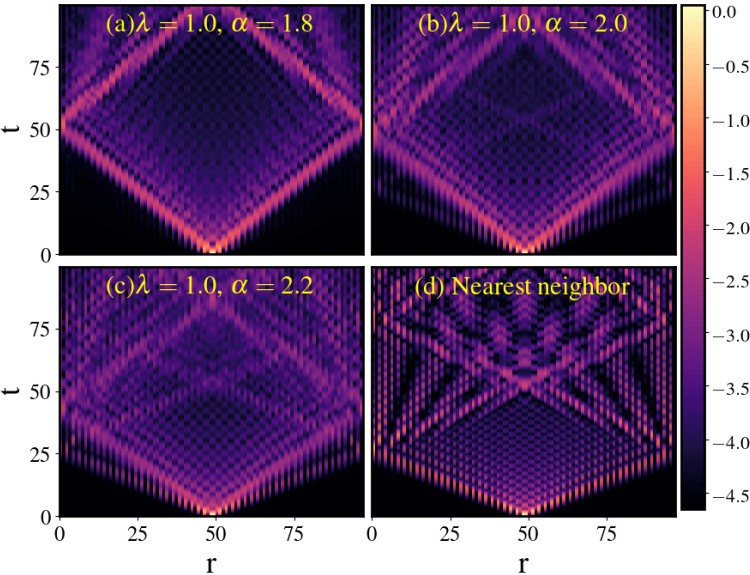

Figure 8: Spinon density $\langle N_\uparrow(r,t)N_\uparrow(r+1,t)\rangle$ for $\lambda = 1$ and different values of $\alpha$, obtained using Villain's two-spinon approximation. Results for the nearest neighbor case are also included. Color density is in a log scale.

teristic velocity is an ill-defined problem, since spins are allowed to hop to all distances. This is also reflected in the magnon dispersion no longer having a linear dispersion, as previously observed.

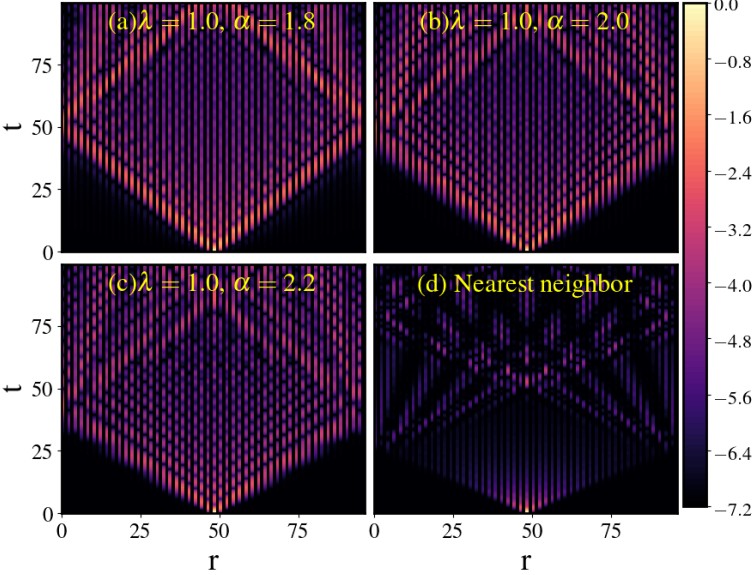

Figure 9: Same as Fig.8 but for the "magnon density" $\langle N_\uparrow(r,t)N_\uparrow(r+1,t)N_\uparrow(r+2,t)\rangle$.

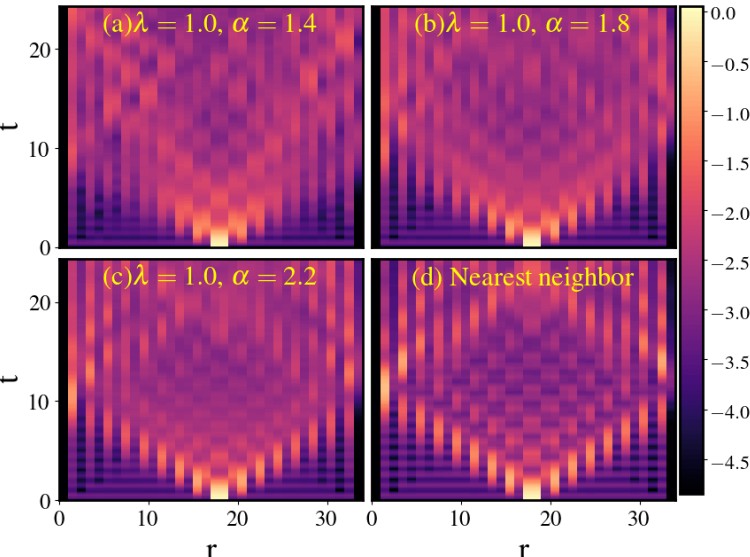

Figure 10: Spinon density $\langle N_\uparrow(r,t)N_\uparrow(r+1,t)\rangle$ for $\lambda = 1$ and different values of $\alpha$, obtained using Villain's approximation including three spinons and one anti-spinon.

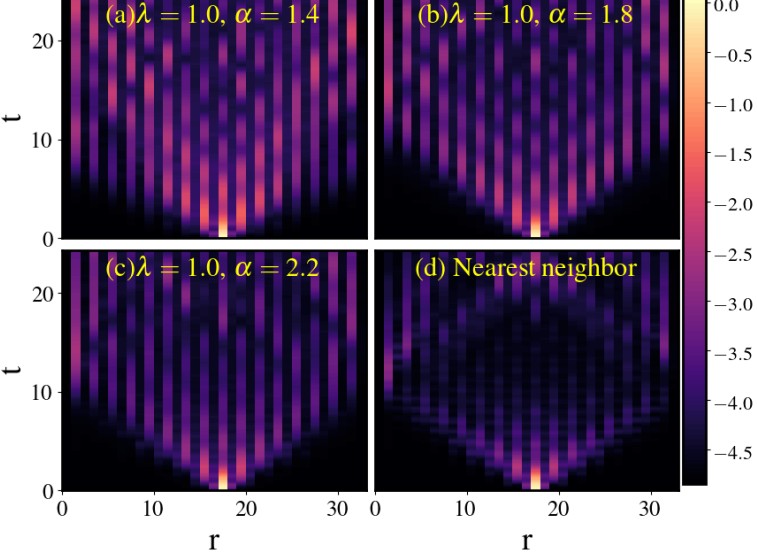

Figure 11: Magnon density $\langle N_\uparrow(r,t)N_\uparrow(r+1,t)N_\uparrow(r+2,t)\rangle$ for $\lambda = 1$ and different values of $\alpha$, obtained using Villain's approximation including three spinons and one anti-spinon.

## 5 Summary and conclusions

We have analyzed the spectrum and studied the nature of the excitations of a Heisenberg chain with staggered long range interactions. The unfrustrated long-range nature of the exchange effectively increases the dimensionality of the system and the chain is able to undergo true symmetry breaking and develop long range order. For weakly decaying interactions, our

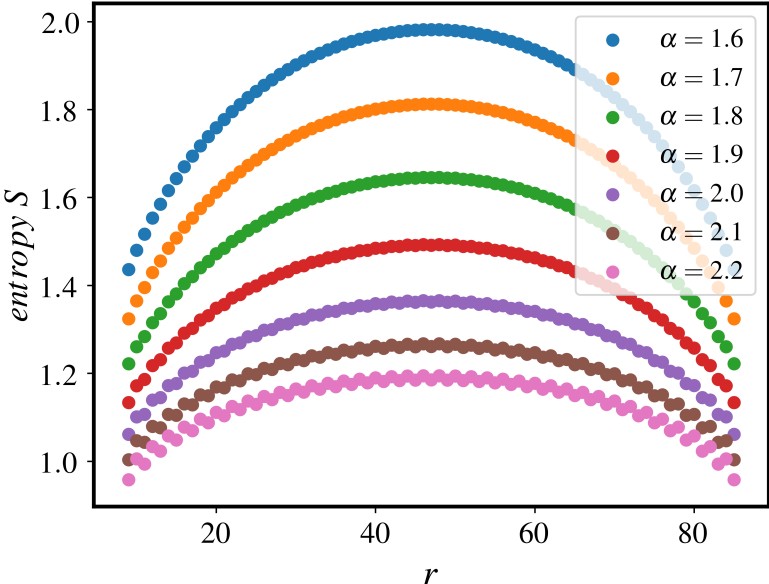

Figure 12: Von Neumann entanglement entropy calculated for a chain with $L = 96$ sites across the quantum critical point.

tDMRG calculations show that the emergence of Néel order can be associated to the formation of bound states of spinons that become coherent quasiparticles (magnons). At the same time, the two-spinon continuum is pushed to higher energies. This is supported by two-spinon and three-spinon approximations that reproduce the main features and explain the formation of bound states due to a confining potential that grows logarithmicaly. The observed behavior bears very close resemblance to the one found in actual neutron experiments in higher dimensional materials. The apparent super-ballistic behavior observed in the time-dependent correlations can be identified with spinons "leaking out" of the lightcone, as observed in the quantum Ising model [54, 55] and the ferromagnetic Heisenberg model with power-law interactions [50, 56, 57]. Our calculations within the generalized Villain approximation support similar conclusions, in agreement with the results in those models: the sublinearity of the dispersion is associated to multiple quasi-particles propagating at different velocities [58–63]. Interestingly, while the magnon dispersion is linear in two dimensions, a rather similar effect occurs in which the momentum dependence of the group velocity gives rise to wavepackets propagating ballistically but with different velocities in different directions [64].

**Funding information**    The authors acknowledge support from the National Science Foundation under grant No. DMR-1807814.

## A    Ground-state calculations

Since the computational cost of DMRG is associated with the entanglement area law, it is expected that the all-to-all interactions will make the calculations considerably more challenging. As seen in Fig.12, the Von Neumann entanglement entropy grows and practically doubles in the narrow window across the transition from disordered to Néel. This occurs as a smooth crossover, and the entanglement entropy and its derivatives vary continuously across the quantum critical point. We also studied the behavior of the entanglement spectrum (not shown), and found that the structure of the "tower of states" changes from singlet-triplet-triplet in the

disordered phase to singlet-triplet-quintuplet in the symmetry broken one. Notice that, even though the system remains critical throughout the entire range of parameters, the problem cannot be described in terms of a conformal field theory and, therefore, analytical predictions are not possible. However, the larger entanglement in the Néel phase reflects the fact that the long-range interactions effectively increase the dimensionality of the problem, and the area law becomes a volume law. Curiously, reliable results with a truncation error of $10^{-6}$ are achievable with 600-800 DMRG basis states.

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
