# Peer review of "From deconfined spinons to coherent magnons in an antiferromagnetic Heisenberg chain with long range interactions"

_SciPost Physics, doi:SciPost Phys. 10, 110 (2021)_

## Round 1 · Referee Report · Anonymous (Referee 1) · 2020-9-22

Strengths

1- Timely and interesting topic. 2-Clear text and statement of the main results which appear to me to satisfy the criteria of SciPost Physics. 3- Interesting and intuitive explanation of the low energy excitation spectrum of the model.

Weaknesses

1- Some figures are not clear and need some reordering. 2- The interpretation of some numerical results needs to be clarified.

Report

The present manuscript presents a picture of the excitation spectrum of an Heisenberg chain with long range anti-ferromagnetic interactions (no frustration) across the transition as a function of the power law decay exponent between Neel and disordered phase.

The main physical picture that emerges in this analysis is the transition is accompanied by the binding of pairs spin on excitations into Magnons as seen in the spin structure factor and studying the dynamics after a spin flip.

I find the results interesting and meeting the criteria for SciPost Physics (up to my knowledge of the literature on the subject). I find however the analysis of some plots rather simplistic and I would like the authors to revisit more critically some of their claims.

Requested changes

1- The splitting of the magnon bound state from the continuum is quite visible in Fig.2 but not in Fig.3 were one sees a peak of more or less the same width with some changing features at higher frequencies. The unchanging width is probably due to the finite and rather small t_{max} that can be achieved by tDMRG that makes it hard to discriminate between a sharp peak and the edge of a continuum. I would comment on this in the caption and try, if possible, to present Fig.3 in a clearer way (just two curves for small and larger alpha are enough, perhaps in two separate panels). I am wondering if relaxing a bit the requirements of accuracy of tDMRG and getting to longer times would allow to get a sense of the trend which would make this figure much more convincing.

2- At some point it is stated “ It is reasonable to expect that as more spinon and anti-spinon states are included, only one band will finally survive“ without any discussion. If the multiple bound states are just bound states of the confining potential in Fig.1(b) does adding a spinon and anti spinon make the higher ones decay ? What is the mechanism at play ?

3- Nonlinearities such as the ones observed in panel (a) and (b) of Fig.(7) have been seen in other problems (see Hauke and Tagliacozzo (Prl) for the quantum Ising chain). There they are attributed simply to the fact that for alpha<2 (for the Ising chain) the system displays true long range properties and in particular power law decaying correlations in space. In this manuscript it is stated that the non linearity can be attributed to the spinon, anti-spinon contributions. I believe it would be important for the authors to comment on this and clarify the issue.

  • validity: high
  • significance: good
  • originality: good
  • clarity: top
  • formatting: excellent
  • grammar: perfect

Author:  Adrian Feiguin  on 2020-12-18  [id 1085]

(in reply to Report 1 on 2020-09-22)
Category:
answer to question
reply to objection
pointer to related literature

We thank the Referee for his/her positive assessment of our work and his/her comments and suggestions. We address them in the following, hoping that the manuscript can be accepted for publication:

1- We welcome the Referee's suggestion. We have discussed this at length and we concluded that the current plot is the one that better describes the merging of the continuum with the quasi-particle peak. I suppose this is on the eye of the beholder, and we have a biased perspective, but we hope that the Referee can accept our point of view. As a matter of fact, we have tried his/her suggestion and simulated longer times and even longer systems keeping less states (this is why our re-submission took so long). Unfortunately, the results are not acceptable, displaying considerable artifacts and spurious oscillations at low energies. We need to settle for the previous data, as the method's limitations with long range interactions and the growth of the entanglement make the calculations very challenging. Although limited in resolution, we trust that the current results are well converged. We have included a comment in the caption of Fig. 3 indicating that "The finite width of the peaks is determined by the maximum simulation time. "

2- We agree that this point is not clear in the text. We hope that the following sentence clarifies this point: "It is reasonable to expect that as more spinon and anti-spinon states are included, more states will appear at low energies. The fact that only one band finally survives would indicate that higher energy bound states (magnons) tend to decay into the continuum of multi-spinon excitations."

3- We thank the Referee for bringing this paper to our attention. We have added the corresponding reference, as well as one for Frerot2018. The new text, included in the conclusions, reads: "The apparent super-ballistic behavior observed in the time-dependent correlations can be identified with spinons ``leaking out'' of the lightcone, as observed in the quantum Ising model\cite{Hauke2013} and the ferromagnetic Heisenberg model with power-law interactions\cite{Frerot2018}. Our calculations within the generalized Villain approximation support similar conclusions, in agreement with the results in those models: the sublinearity of the dispersion is associated to multiple quasi-particles propagating at different velocities. Interestingly, while the magnon dispersion is linear in two dimensions, a rather similar effect occurs in which the momentum dependence of the group velocity gives rise to wavepackets propagating ballistically but with different velocities in different directions\cite{Wrzosek2020}. "

---

## Round 1 · Referee Report · Anonymous (Referee 2) · 2021-1-4

Strengths

1- Addressing the problem of this unconventional quantum phase transition from dynamical quantities 2-Strong numerics 3-Possible relevance to cold atom experiments

Weaknesses

1- Absence of quantitative estimator for the quantum critical point \alpha_c using dynamical response 2-Absence of discussions on the numerical performance of DMRG for this long-range problem, including for static quantities, such as the entanglement entropy

Report

In this numerical work, the authors study the quite interesting problem of the long-range ordering transition observed in an antiferromagnetic Heisenberg S=1/2 chain with unfrustrated power-law decaying couplings.

They very carefully address the qualitative change in the dynamical structure factor across this unconventional phase transition between (i) a Luttinger liquid regime with deconfined spinon excitations and (ii) a N\'eel ordered state with magnons which are seen as 2-spinon bound states.

Their tDMRG simulations provides a very clear picture supporting a transition around \alpha=2.2. They also provide an intuitive interpretation based on a classical description using a variational Villain-like approach.

The work is clearly written, and easy to follow. The results are interesting and also lead to potential experimental checks for time of flight experiments (if staggered Heisenberg long-range coupling are achievable in cold atoms...)

I don't have strong objections or negative comments to make, but only a couple of suggestions to make in order maybe to improve some parts.

0/ In Fig.1, the confining potential V(r) is not formally defined. 1/ It could be good for the DMRG community interested in long-range coupled systems to briefly discuss the performances of the technique when alpha is decreased. Perhaps also showing the entanglement entropy would have been interesting. 2/ Would it be possible to build a quantitative estimator for the transition based on the spectral response?

  • validity: high
  • significance: high
  • originality: good
  • clarity: high
  • formatting: perfect
  • grammar: perfect

Author:  Adrian Feiguin  on 2021-01-13  [id 1151]

(in reply to Report 2 on 2021-01-04)

We thank the Referee for his/her positive assessment of our manuscript and the insightful comments. We have addressed his/her concerns in the new version of the manuscript as follows:

0/ In Fig.1, the confining potential V(r) is not formally defined.

Answer: We have included the definition explicitly (basically, the energy cost of separating two someone walls/spinons in the Ising limit of the model)

1/ It could be good for the DMRG community interested in long-range coupled systems to briefly discuss the performances of the technique when alpha is decreased. Perhaps also showing the entanglement entropy would have been interesting.

Answer: We have included an Appendix with more information about the ground-state calculations, including a new Fig. 12 with the entanglement entropy as a function of the parameter alpha.

2/ Would it be possible to build a quantitative estimator for the transition based on the spectral response?

Answer: It is very difficult to pinpoint the exact point at which the two spinon continuum merges with the bound state/magnon band. However, we found that the entanglement spectrum changes qualitatively on both sides of the transition: the tower of states reflects a sequence singlet-triplet-triplet-quintuplet in the disordered phase, while it is singlet-triplet-quintuplet in the symmetry broken one. This could potentially be used to identify the quantum critical point, even though the entanglement entropy behaves smoothly through the transition. We added this information in the appendix.

---

## Round 2 · Referee Report · Anonymous (Referee 1) · 2021-3-17

Report

Dear editor, I have read in detail the reply by the authors and I am fully satisfied. The paper can be published as it is.

---

## Editorial Decision

published